# Enhancement of Soybean Meal Alters Gut Microbiome and Influences Behavior of Farmed Atlantic Salmon (*Salmo salar*)

**DOI:** 10.3390/ani13162591

**Published:** 2023-08-11

**Authors:** Alexandra Leeper, Clara Sauphar, Benoit Berlizot, Gabrielle Ladurée, Wolfgang Koppe, Stephen Knobloch, Sigurlaug Skírnisdóttir, Rannveig Björnsdóttir, Margareth Øverland, David Benhaïm

**Affiliations:** 1Department of Research and Innovation, Iceland Ocean Cluster, Grandagardur 16, 101 Reykjavik, Iceland; 2Department of Animal and Aquaculture Sciences, Faculty of Biosciences, Norwegian University of Life Sciences, 1420 Aas, Norway; 3Department of Research and Innovation, Matís Ltd., 12, Vínlandsleid, 113 Reykjavik, Iceland; 4Department of Aquaculture and Fish Biology, Hólar University, 551 Hólar, Iceland; 5Department of Biological Sciences Ålesund, Norwegian University of Science and Technology, 6025 Ålesund, Norway; 6Department of Food Technology, Fulda University of Applied Sciences, 36037 Fulda, Germany; 7Faculty of Natural Resource Sciences, University of Akureyi, Nordurslod, 600 Akureyi, Iceland

**Keywords:** soybean, coping style, welfare, prebiotic, non-starch polysaccharides

## Abstract

**Simple Summary:**

The aquaculture sector relies heavily on soybean meal (SBM) and soy-derived proteins, largely due to their availability, low price and favorable amino acid profile. However, for Atlantic salmon, the inclusion of soybean meal, and soy protein concentrate (SPC) in certain combinations has been associated with impacts on gut health and welfare. This study evaluated two SBM treatments that target improved gut health and were formulated for inclusion in freshwater phase salmon diets: enzyme pre-treatment (ETS), and addition of fructose oligosaccharide (USP). These were compared with untreated soybean meal (US) and fish meal (FM). The effects on growth performance, gut microbiome, and behaviors relevant to welfare were investigated. Both diets containing the treated SBM supported growth performance comparable with FM and altered the gut microbiome. Fish fed SBM displayed a tendency toward more reactive behavior compared with those fed the FM-based control. All fish tested had a low response to elicited stress, although ETS-fed fish responded more actively than those fed the US diet. SBM-fed fish had the lowest repeatability of behavior, which may have implications for welfare. Both treatments of SBM are a promising option to optimize the application of this widely used protein source for aquaculture feeds.

**Abstract:**

Atlantic salmon (*Salmo salar*) is one of the worlds most domesticated fish. As production volumes increase, access to high quality and sustainable protein sources for formulated feeds of this carnivorous fish is required. Soybean meal (SBM) and soy-derived proteins are the dominant protein sources in commercial aquafeeds due to their low-cost, availability and favorable amino acid profile. However, for Atlantic salmon, the inclusion of soybean meal (SBM), and soy protein concentrate (SPC) in certain combinations can impact gut health, which has consequences for immunity and welfare, limiting the use of soy products in salmonid feeds. This study sought to address this challenge by evaluating two gut health-targeted enhancements of SBM for inclusion in freshwater phase salmon diets: enzyme pre-treatment (ETS), and addition of fructose oligosaccharide (USP). These were compared with untreated soybean meal (US) and fish meal (FM). This study took a multi-disciplinary approach, investigating the effect on growth performance, gut microbiome, and behaviors relevant to welfare in aquaculture. This study suggests that both enhancements of SBM provide benefits for growth performance compared with conventional SBM. Both SBM treatments altered fish gut microbiomes and in the case of ETS, increased the presence of the lactic acid bacteria *Enterococcus*. For the first time, the effects of marine protein sources and plant protein sources on the coping style of salmon were demonstrated. Fish fed SBM showed a tendency for more reactive behavior compared with those fed the FM-based control. All fish had a similar low response to elicited stress, although ETS-fed fish responded more actively than US-fed fish for a single swimming measure. Furthermore, SBM-fed fish displayed lower repeatability of behavior, which may indicate diminished welfare for intensively farmed fish. The implications of these findings for commercial salmonid aquaculture are discussed.

## 1. Introduction

Atlantic salmon (*Salmo salar*) is one of the most intensively farmed finfish in the world and production volumes continue to grow with consumer demand and rising global population [1]. Atlantic salmon is a carnivore and has a high dietary protein demand [2]. Sourcing protein for inclusion in salmon aquafeeds has become a major economic and sustainability challenge [3]. Historically, fish meal dominated the aquafeed market, but increasing and unsustainable pressure on wild-capture fisheries and rising prices have resulted in the commercial uptake of soy protein sources [3,4]. Soybean meal (SBM) gained initial popularity because of its availability, low cost and a favorable amino acid profile for fish nutrition [5]. However, for Atlantic salmon, SBM impacts gut health, and triggers soybean meal induced enteritis (SBMIE) attributed to the presence of saponins, non-starch polysaccharides (NSP) and other anti-nutritional factors (ANFs) [6,7,8,9,10]. These ANFs can adversely alter the distal gut morphology [11,12], reduce nutrient absorption and digestion [13], and adversely alter the gut microbiome [14], compromising the immune resilience and welfare of Atlantic salmon [15,16]. Consequently, the salmon industry now primarily uses soy protein concentrate (SPC), an alcohol-extracted soy product that reduces the inflammatory effect of SBM. However, impacts on gut health and the gut microbiome of Atlantic salmon have also been observed for SPC [17,18,19]. It is necessary to optimize the treatment of SBM for improved salmon production and welfare [20].

The fish gut microbiome has been linked to key traditional measures of aquaculture productivity, including nutrient digestibility and availability, growth performance [21], metabolism, immune development, and disease resistance [22,23,24,25,26,27]. The composition of the gut microbiome is, in turn, influenced by the feed, host environment [28], and genetics [29,30]. In juvenile salmonids, the gut microbiome is particularly malleable to alterations in dietary protein source [31,32], which influence the establishment and development of the adult gut microbiome [33]. When salmonids are fed diets containing SBM or SPC, the composition of the gut microbiome is distinct from fish fed marine diets. There is a greater presence of bacteria associated with inflammation and imbalances in the community, which have been linked to poor intestinal health of SBM- and SPC-fed fish [17,18]. Increasingly, research is highlighting the potential of dietary treatments that alter the gut microbiome to create functional benefits for farmed fish. Such treatments include the application of probiotics, prebiotics, symbiotics, fermentation and enzyme treatment of ingredients [34,35,36,37,38]. Prebiotics are non-digestible fiber that are not directly used by the host but support the proliferation of desirable gut microbiota [39] and they are widely used in human foods and terrestrial farming. Prebiotics have high consumer and industry acceptance [40]. In aquaculture, fructose oligosaccharides (FOS), and mannan oligosaccharides (MOS) have been used in feeds to enhance the gut microbiome and can even improve growth performance [41,42]. Enzyme treatment has also become a common processing method to improve the value of proteins for animal and aquaculture feeds [35]. Enzyme treatment of plant proteins such as SBM has been used to breakdown long-chain carbohydrates such as NSPs to improve their nutritional value [37,43]. This has the potential untapped benefit of producing shorter chain oligosaccharides, which in fish diets, are not utilized by the host but may have an added prebiotic function [44].

A bi-directional link between the gut microbiome and the central nervous system of animals including fish, has been observed. The link is known as the gut–brain axis [45,46]. In fish, there is evidence that the gut microbiome influences swimming behavior [47], feeding behavior [21] and social behavior [48], which are all non-invasive indicators of welfare. [49]. Since existing research indicates differing diets alter the gut microbiome, it is, therefore, valuable to monitor behavior and the gut microbiome when assessing new diets and managing the welfare of farmed fish [50,51,52].

Five behavioral types relevant to welfare in aquaculture have been identified along five major axes: boldness–shyness, exploration–avoidance, sociability, activity, and aggression [53,54]. When individual behaviors are consistent over time and context, they can be referred to as personality [55,56]. Boldness is a measure of the predisposition to take risks [57], and individuals can be categorized on a continuum from bold to shy. Exploration is a measure of predisposition to engage with a novel environment or object [49], and individuals can be categorized on a continuum from active explorers to cautious explorers [58]. Coping style, closely related to personality, refers to a coherent set of behavioral and physiological stress responses, which is consistent over time and which is characteristic to a certain group of individuals [55]. Individuals of a given population can be classified as reactive or proactive. Proactive fish are bold individuals that take more risks and explore their environment faster (less cautiously) when exposed to novelty [59,60]. They are more aggressive [60,61] and display low variability and flexibility in their behavior to changes in their environment [62]. Conversely, reactive fish are shyer, tending to be risk-averse and neophobic. They show greater behavioral flexibility, are more responsive to changes in their environment [63], and are more sociable than proactive individuals [64].

When behavioral traits are correlated, this indicates a behavioral syndrome. Behavioral syndromes can have consequences for welfare, particularly indicating if an individual has the freedom to express natural behavior [51,54,65]. Behavioral traits can be altered by dietary ingredients, making them important indicators to monitor [15,47,66]. In juvenile Senegalese sole (*Solea senegalensis*), replacing fish oil with vegetable oil resulted in significantly more reactive individuals [67]. It is important to understand how dietary treatments affect the coping style and stress responses of Atlantic salmon since they could have impacts on their productivity, fitness and welfare [53]. 

The objective of this study was to optimize SBM in farmed Atlantic salmon diets and to determine if two novel treatments of SBM (enzyme pre-treatment and addition of prebiotic) supported desirable changes in growth performance, the gut microbiome, and behavior. This study addresses multiple and intersecting knowledge gaps for aquaculture using a relevant model organism in a cross-disciplinary approach.

## 2. Materials and Methods

### 2.1. Experimental Animals and Study Design

Atlantic salmon (*Salmo salar*) were sourced from Stofnfiskur Ltd. (Vogar, Iceland), where they were incubated at 5.5 °C, and eyed eggs were transferred to Laxar ehf. (Kópavogur, Iceland). The fish were raised to first feeding using standard commercial techniques and a start-feed diet BioMar Inicio-plus (Grangemouth, Scotland, UK) of 0.5 mm pellet size at 12 °C water temperature. The fry were transferred to Verid aquaculture station of Hólar University (Saudárkrókur, Iceland) where they were acclimated for 6 weeks prior to the start of the experiment, and feeding continued on BioMar Incio-plus (Grangemouth, Scotland, UK) with combination of 0.5 mm and 1 mm pellet sizes. All the fish were individually weighed and measured under anesthetic (2-phenyoxyethanol at 300 ppm) following a 24 h fasting period and according to the approved practices of the Hólar University animal welfare committee informed by experts trained in animal handling. The fish were split into 12 identical 20 L white cylindrical PVC tanks in a flow-through system, in which the water renewal rate was 100–400% per hour. The water temperature was maintained at 11 ± 1.9 °C under continuous light of 250 ± 50 lux and with 90 ± 10% oxygen saturation. The pH was 7, and the level of total ammonia nitrogen was undetectable. All diets were tested in triplicate. Each tank contained 40 fish with initial weights of 2.2 ± 0.4 g. The experimental feeding period lasted 70 days. The experiment was performed following Icelandic guidelines and within the permits and licenses of Verid aquaculture station.

### 2.2. Experimental Diets and Feeding

Four diets containing four different protein sources were tested. A fish meal-based control (FM) diet, an untreated soybean meal (US) diet, an enzyme-treated soybean meal (ETS) diet, and a diet of untreated soybean meal with added fructose oligosaccharide (USP) were used. The chemical composition of the protein sources used in this study is presented in Table 1. For each of the soy diets (US, ETS, USP), the protein source was included at a 25% level, replacing fish meal. The formulated diets and their nutritional compositions are presented in Table 2. The enzyme-treated soybean for the ETS diet was produced by treatment with a 50:50 blend of Hostazyme X (main activity: Endo-1,4-β-xylanase, side enzyme activity: End-1,4-β-glucanase (cellulase), endo-1,3(4)-β-glucanase, α-amylase, protease) and Hostazyme C (main activity; End-1,4-β-glucanase, side enzyme activity: Endo-1,4-β-xylanase, endo-1,3(4)-β-glucanase, α-amylase). The enzyme preparation was blended in 50 °C water. The solution was added to 2 L of water and 1 kg of soybean meal and mixed for 15 min. The mix was then incubated for 3 h at 50 °C and manually mixed every 30 min. Following incubation, excess water was removed through pressing. The material was then further dried at 80 °C to <10% moisture content. The diets were produced by cold pelletization at Matís Aquaculture Research Station (Iceland). All the dry ingredients were milled to homogenize the particle size (IPHARMACHINE, Germany). The dry ingredients were homogenized in a standard food mixer (KitchenAid, Benton Harbour, ME, USA) and milled to further improve the homogeneity of the feed. The dry mix was returned to the food mixer and fish oil was added, together with 200 mL of water, to produce the optimal consistency for processing in a pasta machine set to 0.5 mm strings (ADE, Hamburg, Germany). The strings were dried in a commercial food dryer (Kreuzmayr, Wallern an der Trattnach, Austria) to <10% moisture content, and then finally milled again to produce pellets between 0.5 mm and 1 mm. During the 70-day feeding trial, the fish were fed continuously by electric belt-feed with identical feed volumes at 15% excess fed based on the feed requirements for this developmental stage [2].

### 2.3. Growth Performance

Fish were fasted for 24 h prior to measuring. After 70-days of feeding, all individual fish were anesthetized (2-phenoxyethanol 300 ppm), according to the Hólar University animal welfare committee standards. The fish wet weight (g) and total length (cm) were measured. The specific growth rate (SGR) (%) over the study period was calculated as follows: SGR = ((Ln(Final Weight) − Ln(Initial Weight)) × 100)/*t*, where *t* is the number of days of the experimental period. Mortality was monitored daily during the trial period.

### 2.4. Gut Sampling

A 7-day recovery period was applied following the final measurements of growth performance. The fish were maintained on their respective experimental diets. The fish were then fasted for 12 h to standardize the time since last feed. After this 12 h fasting period, the gut contents remaining were present in both the mid and distal gut; therefore, the samples were of the luminal- and mucosal-associated bacteria. Three fish per tank *(n* = 9 per diet) were randomly selected for gut microbiome analysis. The fish were humanely euthanised with a lethal dose of anesthetic (2-phenoxyethanol 600 ppm), according to Hólar University animal welfare committee standards. The external surface of the fish was rinsed in 90% ethanol followed by sterile distilled water. The gastro-intestinal (GI) tract and contents (from the start of the mid-gut, just below the pyloric caeca to the end of the distal gut) was sampled under sterile conditions and directly frozen at −80 °C.

### 2.5. DNA Extraction, PCR Amplification and Sequencing

Each individual GI tract was homogenized in a sterile petri dish with a sterile scalpel to physically break up the material. The sample was transferred to a sterile 2 mL Eppendorf tube with 300 µL of sterile 1 mm diameter silica beads (BioSpec Products, Bartlesville, OK, USA) and 800 µL of CD1 solution from the QIAamp PowerFecal Pro DNA kit (QIAGEN, Hilden, Germany). The samples were vortexed for 5 s and shaken at maximum speed (30 Hz) in a laboratory mixer mill (Retsch MM400) for 1 min. The supernatant (~800 µL) was transferred to the PowerBead Pro tube from the QIAGEN QIAamp PowerFecal Pro DNA kit. The protocol for this DNA extraction kit was then followed and the final DNA was eluted with 80 µL of C6 solution. A DNA negative (no material added) was run parallel to ensure no contamination occurred during DNA extraction. The DNA concentration was measured for 2 µL of each sample with the Invitrogen Qubit dsDNA BR Assay kit (Invitrogen, Carlsbad, CA, USA). The DNA was diluted to 4 ng µL^−1^ in 50 µL aliquots. The samples were subjected to PCR of the V3-V4 regions of the 16S rRNA gene with a universal bacterial primer pair S-D-Bact-0341-b-S-17 (5′-CCTACGGGNGGCWGCAG-3′)/S-D-Bact-0785-a-A-21(5′-GACT-ACHVGGGTATCTAATCC-3′) [57]. The PCR master mix included diluted DNA, nuclease-free water, Q5 High Fidelity DNA polymerase (New England Biolabs, Ipswich, MA, USA), Q5 GC Enhancer, 0.5 µM of each primer containing Illumina overhand adapters, 1 × Q5 Reaction buffer, and 200 µM dNTPs (New England Biolabs, Ipswich, MA, USA). Both positive and negative samples were also run through the PCR to monitor for absence of contamination and successful amplification. The thermocycling protocol had an initial denaturation step of 90 °C for 30 s, followed by 35 cycles of denaturation (90 °C for 10 s), annealing (52 °C for 30 s), and an extension step (72 °C for 30 s), with a final extension (72 °C for 2 min). The libraries were multiplexed with Nextera XT v2 barcodes (Illumina, San Diego, CA, USA), normalized using Sequel-Prep normalization plates (ThermoFisher Scientific, Waltham, MA, USA) and sequenced on a MiSeq desktop sequencer (Illumina, USA) using v3 chemistry and 2 × 300 cycles.

### 2.6. Behavior

Eight fish from the remaining tank population were randomly selected to be uniquely tagged with Visible Implant Elastomer (VIE) color tags (Leblanc and Noakes, 2012) under anesthetic (2-phenoxyethanol 300 ppm). Each fish was injected parallel to the dorsal fin, below the surface skin layer with two 1 cm strips of color to uniquely mark each of the 8 selected fish per tank (n = 24 per diet). All remaining untagged fish per tank were retained in the tank to maintain a consistent stocking density but were not included in the behavioral observations. Each tagged fish was observed in two different behavior contexts (a swimming test and an open field test with shelter), and each of these tests was performed two times. Between each test, there was a one-week recovery period, during which the feeding practices and respective diets were maintained. The behavior observations were performed in a dedicated room to minimize external disturbance. Fish were individually tested for all behavioral observations. Behavior was recorded using a monochrome camera (Basler Ace acA1920-150 um, Germany) with a frame rate of 30 Hz and resolution of 1280 × 1024 pixels. The camera was placed 112 cm (swimming test) and 110 cm (open-field test) above the water level of the respective behavioral apparatus. The video recordings were analyzed with EthoVision XT 14 software (Noldus, The Netherlands), which tracked fish movement. The VIE tag markings were recorded after each behavior observation so individual data could be tracked across both time and context.

#### 2.6.1. Swimming Test

A swimming test was used to assess the data for both the stress response and the exploration–avoidance axis. The apparatus consisted of 4 circular arenas (diameter = 25 cm, water depth 7.8 cm (4 L), height = 15 cm). The arenas were illuminated from below to provide uniform light intensity (260 lux) (see Figure 1). Tagged fish were collected from their respective tank and transferred in identical white transfer buckets with closed tops and then randomly placed into numbered single fish arenas. All four arenas were filmed simultaneously and continuously for 20 min. Four distinct periods were designated. The 0–5 min period was the acclimation period (AC), and 5–10 min was the normal swimming period (NS). At the 10 min mark, a uniform stress was elicited (the bottom lighting was switched off for 3 s and then switched back on again). The 10–15 min period was the post light stress period (PLS), and 15–20 min was the recovery period (RC). Behavior was not recorded while the light was off.

The following variables were collected: mean distance from the center-point (cm)—the distance of the center-point of the fish body from the arena center (DisCent); total distance moved (cm)—the distance travelled by a fish measured from the center-point of the body between consecutive X-Y coordinates (TotDis); mean velocity (BL s^−1^)—the distance moved by the center-point of the body per unit time between consecutive X-Y coordinates normalized to body lengths per second (Vel); and absolute angular velocity (° s^−1^)—calculated by EthoVision XT software as Vangn = RTAn/tn − (tn − 1), where RTAn represents the relative turn angle for sample n, and tn − (tn − 1) is the time difference between the current and previous sample (AngVel). Finally, mobility state—the cumulative duration in which the fish body is changing—was assessed at three levels: highly mobile (s) when cumulative duration is 60% (HiMob); moderate mobile (s) when the cumulative duration is between 20 and 60% (MedMob); and immobile (s) when cumulative duration is below 20% (Immob).

#### 2.6.2. Open-Field Test

An open-field test (OFT) with a shelter was used to assess boldness along the bold–shy behavior axis [68,69,70]. The apparatus consisted of 4 rectangular arenas (40 × 30 × 25 cm), with a water depth of 6 cm (7 L) and an identical shelter placed in the bottom-right of each arena (14 × 6.5 × 6.5 cm). The arenas were illuminated from below to provide uniform light intensity (260 lux) (see Figure 2A). The tagged fish were collected from their respective tank and transferred in identical white transfer buckets with closed tops and then randomly placed in numbered single fish arenas, directly into a shelter. The fish were placed in the shelter through a top compartment (4 cm diameter) which was then closed. The main shelter door (a sliding opaque trapdoor) was kept shut, closing off access to the arena. All four arenas were filmed simultaneously and continuously for 25 min. During the first 5 min, the fish remained shut inside the shelter for the acclimation period, after which the door to each arena shelter was lifted simultaneously and filming continued for a further 20 min uninterrupted. Each arena was virtually divided into four zones using the EthoVision XT software. These zones were called Shelter, Entry, Border and Center (see Figure 2B). The center zone is considered high-risk and staying close to the border edges of a space is considered more cautious and an indicator of shy behavior [57,71]. Variables used to characterize OFT behavior for the trait of boldness were as follows: Latency (s) to emerge—time taken to exit the shelter (Lat); time spent in each zone (s) (Shelter, Entry, Border, Center); mean distance from the shelter (cm) (DisShelt); number of returns to shelter (Ret); cumulative duration highly mobile (s) (HiMob); distance moved (cm)—the distance travelled by a fish measured from the center-point of body between consecutive X-Y coordinates (TotDis); mean velocity—the distance moved by the center-point of body per unit time between consecutive X-Y coordinates normalized to body lengths per second (BL s^−1^); and absolute angular velocity (° s^−1^) (AngVel) calculated as in the swimming test.

### 2.7. Statistical Methods

Statistical analyses were performed in Rstudio version 3.6.1 (05.08.2019). Tests for growth performance, gut microbiome and behavioral characteristics were two-tailed, with a significance level set to α = 0.05. Multivariate linear mixed models were fitted to estimate the repeatability and correlation between the personality estimates using Bayesian statistical modelling.

#### 2.7.1. Growth Performance

SGR (%) was used as a proxy for growth performance across the different diets. A Linear Mixed Model (LMM) with the package lme4 [72] was selected, defining diet (FM, US, ETS, USP) as a fixed factor and tank as a random nested factor of feed diet. Since the random nested factor of tank did not cause significant variation in SGR (%), a simplified statistical test was adopted. A Linear Model (LM) with the package nmle [73] was used, where diet was a fixed factor and the model residuals had a normal distribution. A Tukey post hoc test was applied to assess pairwise differences between diets.

#### 2.7.2. Gut Microbiome

The gut microbiome was analyzed from demultiplexed FASTQ files from Illumina which were processed to produce amplicon sequence variants (ASVs) with DADA2 package version 1.16.0 [74]. The filterAndTrim variables were as follows: truncLen = c(280,250), trimLeft = 21, maxN = 0, maxEE = c(2,2), and truncQ = 2. Taxonomy was assigned to ASVs from version 138 of the SILVA database and the function assignTaxonomy [75]. The R packages, phyloseq [76], microbiome [77] and vegan [78] were selected for the analyses, and ggplot2 was selected to visualize the key data [79]. The average read number output from the DADA2 pipeline was 20,643 ± 12,702, and two samples were removed as they did not contain any reads after processing. Five PCR negative samples were also sequenced to control for any contamination that may have occurred during sample amplification. These controls were used to remove suspected contamination from samples using the decontam package, with the prevalence method and a threshold of 0.5 [80]. The read depth was normalized across all samples using the function rarefy_even_depth. The raw 16S rRNA gene amplicon reads can be found in the Sequence Read Archive.

Alpha and beta diversity indices were used to quantitatively analyze the gut microbiome. The alpha diversity measures selected were observed richness of AVSs, Shannon diversity, Chao1 diversity and Pielou’s evenness. A GLMM was used to test for significant difference in the alpha diversity measures between diets. In this model, dietary treatment was a fixed factor and tank was a random nested factor of feed treatment (which was tested by a Likelihood Ratio Test [81]). A Tukey test was applied for post hoc testing. Beta diversity was assessed by transforming microbiome community data using a Bray–Curtis dissimilarity matrix. An Analysis of Similarity (ANOSIM) was performed to test for significant difference between and within the gut microbiome communities of fish fed the different diets. The relative abundance of taxa at the phylum level and genera level as a proportion was calculated. The taxa present in the community at <1% relative abundance were grouped into a category labelled “Other”.

#### 2.7.3. Behavior

To assess stress response, all the swimming test variables were analyzed using a Linear mixed effect models (LMM). For each analysis, the explanatory variables included in the full model were diet (FM, US, ETS, USP), period (AC, NS, PLS, RC) and fish total length (cm) (TL), and interactions between diet and period were considered. Random effects considered in the model were trial replicate number (1, 2), and tank (1, 2, 3) nested in diet, and individual VIE tag (ID).

To assess the behavior trait of exploration, TotDis values recorded for the AC and NS periods were selected [82]. All the variables measured in the OFT (to assess boldness) were collapsed into first principal component scores using a Principal Components Analysis (PCA). A correlation matrix was used to verify multi-collinearity between variables, i.e., to identify variables that correlate very highly (r = 0.9) with one or more variables, and such variables were removed in the following analyses. The remaining selected variables used for the PCA were TotDis, Ret, Entry, Border, Center, Shelter, and AngVel. Each individual fish was then assigned a score (boldness score) from the component that explained most of the variation in the data.

Both traits, boldness and exploration, were then compared between diet in two respective LMM models where boldness score or TotDis were the response variables. The full models were reduced by backward selection based on the Akaike Information Criterion (AIC) [83]. Diagnostics based on residuals of the model were performed to assess the adequacy of the reduced model and compliance with the underlying assumptions. The dependent variables were transformed whenever necessary to ensure that the residuals followed the assumed error distribution. Finally, the effects of the independent variables were estimated from the reduced models and their significance was tested by likelihood ratio tests (LRT) between models, respecting the marginality of the effects that are supposed to follow a χ^2^ distribution under the null hypothesis (type II tests; [81]). This analysis was followed by a post hoc multiple comparison test [84] to assess pairwise differences between models.

#### 2.7.4. Repeatability and Correlation between Behavioral Traits

A Bayesian multivariate linear model using Stan [85] was run using the brms package [86], with a single model for each diet (four models in total). Each model simultaneously regressed each dependent variable (i.e., boldness score and TotDis) against a set of fixed factors (Tank, Trial, SGR and TL). The random-effects structure included ID as a grouping variable, allowing us to calculate the repeatability of Boldness and TotDis as the ratio of the among individual variance and the sum of the among individual and residual-level variances (i.e., personality) [87]. Fish that did not exit the shelter for at least one of two the trials were removed from the analysis. Moreover, the model estimated covariances between Boldness and TotDis at both the ID and residual levels. The among individual covariance quantified the degree to which Boldness and TotDis was correlated among individuals across multiple trials (i.e., behavioral syndrome). The model was run for 64,000 iterations (32,000 for warm-up and 32,000 for sampling), using four chains; adapt delta was set to 0.9, and the max tree depth to 40. All other parameters were set to default. Convergence was assessed using the standard diagnostics provided by Stan, including the potential scale reduction factor (R), effective sample size, and visual inspection of trace plots and histograms for each model parameter. Unless otherwise noted, posterior modes were used for point estimates and higher posterior density, with 95% coverage for uncertainty intervals (UI95%), calculated using the map_estimate and hdi functions, respectively, from the “coda” package [88].

## 3. Results

### 3.1. Growth Performance

There was a significant difference in SGR% between diets (see Figure 3). Fish fed FM had significantly higher SGR% than those fed US. Fish fed the enhanced soy diets, ETS and USP did not have a significantly different SGR% from either FM- or US-fed fish.

### 3.2. Gut Microbiome

There was a significant difference in the gut microbiome community alpha diversity measures for Shannon diversity (see Figure 4B) and for Pielou’s evenness (see Figure 4D), but not for the observed richness of ASVs (see Figure 4A) or Chao1 diversity (see Figure 4C), between the dietary treatments. For both Shannon diversity and Pielou’s evenness, the highest community diversity and evenness values were observed for the FM diet, which was significantly higher than the values for all other diets. The US diet-fed fish guts had comparable Shannon diversity and evenness to the USP-fed fish. Fish fed the ETS diet had significantly lower gut microbiome Shannon diversity and evenness than fish fed any other diet.

The gut microbiome differed significantly between diets. There was a greater similarity between individual fish with the same diet than between individuals with different diets (ANOSIM *p* = 0.001, R = 0.84). The NMDS plot (see Figure 5) shows distinct clustering by feed treatment.

At the phylum level, for all dietary treatments, the phylum present in the largest proportions was *Firmicutes* (FM = 0.81 ± 0.1, US = 0.93 ± 0.03, ETS = 0.99 ± 0.01, USP = 0.95 ± 0.03), followed by lower proportions of *Actinobacteria* (FM = 0.1 ± 0.06, US = 0.05 ± 0.03, ETS = 0.01 ± 0.01, USP = 0.03 ± 0.01) and *Proteobacteria* (FM = 0.09 ± 0.12, US = 0.01 ± 0.01, ETS = 0.01 ± 0.01, USP = 0.02 ± 0.03), which were at similar levels.

There were nine genera with a relative abundance > 1% of the community (see Figure 6). All other genera were present at very low relative abundances (<1%). The genera composition between fish fed the FM diet and the three diets containing soy showed distinct differences. The genera, *Anaerosalibacter, Clostridium sensu stricto 18*, *Clostridium sensu stricto 7, Hathewaya* and *Peptosteptococcus* all had a greater relative abundance in the guts of fish fed the FM diet than in the guts of fish fed the US, ETS or USP diets. There was a trend that the relative abundance of the gut microbiome was greater in US and USP-fed fish than in the ETS-fed fish. The LABs *Leuconostoc* and *Weissella* had the lowest relative abundance in the FM fish gut microbiomes, slightly higher relative abundance in the ETS fish gut microbiomes, and their relative abundance was greatest in the US- and USP-fed fish. For the LAB *Enterococcus*, there was a differing trend. The relative abundance in the gut microbiome was very high in the gut of ETS-fed fish, low but present in the FM- and USP-fed fish, and absent in the US-fed fish. The *Staphylococcus* presence was comparable between the gut microbiome of fish fed FM, US and USP, but lower in ETS-fed fish.

### 3.3. Behavior

#### 3.3.1. Swimming Test

There were no significant differences in any of the swimming variables between fish fed different diets. There was a significant difference in swimming activity between periods for all variables regardless of diet. However, this did not appear to be driven by the elicited light stress but instead by the amount of time fish spent in the arena. The interaction between diet and period was significant for AngVel (° s^−1^) for USP-fed fish during the PLS period (χ^2^ = 17.53, df = 9, *p* = 0.03). Vel had a significant effect for fish fed the US diet during the RC period (χ^2^ = 17.54, df = 9, *p* = 0.041). There was a significant effect of swimming test replicate (1, 2) for TotDis (cm) and DisCent (cm). For all variables, there was a significant effect of TL (cm) on the variables DisCent (cm), TotDis (cm), and Vel (BL s^−1^), i.e., there was an inverse relationship with TL (cm) (larger fish had lower values for these variables). For AngVel (° s^−1^) there was a positive relationship with TL (cm) (larger fish had higher values for this variables). TotDis during the AC and NS periods (proxy for exploration) did not significantly vary between feed treatments (χ^2^ = 0.46, df = 3, *p* = 0.93) (see Figure 7), but there was a highly significant effect of TL (χ^2^ = 71.4, df = 1, *p* < 0.0001), i.e., the higher the TL, the lower the TotDis.

#### 3.3.2. Open-Field Test

PC1 explained 48% of variation in the data. For PC1, a higher value indicates a greater total distance travelled (cm) (loading = 0.43), higher swimming velocity (BL s^−1^) (loading = 0.41), greater number of returns to the shelter (loading = 0.3), greater time spent in the entry zone (s) (loading = 0.12), greater time spent in the border zone (s) (loading = 0.41), greater time in the center zone (s) (loading = 0.21), and higher mobility (s) (loading = 0.31). A lower value indicates greater time spent in the shelter (s) (loading = −0.35), latency to exit shelter (s) (loading = −0.25), and greater absolute angular velocity (° s^−1^) (loading = −0.21). PC1, therefore, presents a gradient from shyer (low values) to bolder (high values) and was used hereafter as proxy for the trait of boldness. There was no significant difference in boldness between diets, although there was a trend visible showing FM-fed fish to be bolder than fish fed any of the soy diets, and the US-fed fish to be shyer than any other fish (see Figure 8). TL (cm) had a significant effect on boldness (*p* =< 0.001, df = 1, χ^2^ = 31.92, S.E = 0.11, Estimate = −0.6). The random factor test repeat number was not significant, but the random factors individual ID and tank were significant.

#### 3.3.3. Repeatability and Correlation of Behavior Traits

Among individual variances of both boldness and exploration, traits were unambiguously different from zero in fish fed all diets, with the exception of fish fed US and USP diets for the trait of boldness (see Table 3). Indicating individual repeatability in these behaviors. The highest repeatability for the trait of boldness was observed for fish fed the FM diet. The highest repeatability for the trait of exploration was observed for fish fed the USP diet. The large confidence interval in the US-fed fish for the trait of exploration shows a high degree of uncertainty in the repeatability estimate. The repeatability of the exploration trait for ETS-fed fish was also high, but the repeatability of the boldness trait was low for the same diet. The covariance of boldness and exploration for fish fed all diets, except for the fish fed the ETS diet, was close to zero, with UI 95% strongly overlapping zero (see Figure 9). For the ETS-fed fish, neither ID nor residuals overlapped with 0, which indicates a positive correlation between boldness and exploration. For fish fed the FM, ETS and US diets, exploration tended to be greater with lower TL measures. For fish fed FM, exploration tended to be greater with higher SGR %. In fish fed the USP diet, boldness tended to be greater with higher SGR %, and exploration tended to be lower in the second trial.

## 4. Discussion

The objective of the present multi-disciplinary study was to optimize the application of soybean meal (SBM) in formulated feeds for Atlantic salmon (*Salmo salar*). This study investigated two potential novel treatments of SBM in formulated diets, an enzyme pre-treated SBM diet (ETS), and an SBM diet with addition of the prebiotic (fructose oligosaccharides, FOS) (USP). These diets were compared with an untreated SBM diet (US) and a fish meal control diet (FM). The effects on growth performance, gut microbiome, and individual behavior traits relevant to the welfare of cultured salmon are reported.

The growth performance of all Atlantic salmon in this study was within a normal range for the freshwater phase (FW) and for the experiment water temperature [89,90]. The findings were consistent with the existing literature, showing that FM diets support higher growth rates than US for juvenile Atlantic salmon. The existing literature reports this impact at inclusion levels of 20% SBM [12], and at higher levels of 31% SBM [91]. In the seawater phase (SW), inclusion levels of 20% SBM and higher show reduced growth rate in Atlantic salmon compared with those fed FM diets [11], although inconsistencies in the literature exist [92]. In conflicting existing studies, salmon parr fed diets containing 16.7% SBM [93] or 40% SBM [94] performed comparably with those fed FM diets. The two treatments of SBM tested in the present study, ETS and USP, supported growth performance comparable with that of FM, suggesting that both enzyme pre-treatment of SBM and addition of FOS have potential growth-based benefits for improving SBM application in diets of FW salmon. Similar benefits have been seen with SBM treated to reduce the content of oligosaccharides and anti-nutritional factors (ANFs), which improved salmon growth compared with an untreated SBM at a 40% inclusion level [95]. In Japanese seabass (*Lateolabrax japonicus*), non-starch polysaccharides (NSP) and enzyme treatment of feed also had beneficial effects on growth performance [96]. Likewise, in the white-spotted snapper (*Lutjanus stellatus*), NSP-targeted enzyme treatment to *Gracilaria lemaneiformis* supported improved growth [97]. Conversely, other studies found no growth performance benefits for Atlantic salmon fed a comparable NSP-targeted enzyme treatment of SBM [98]. Furthermore, phytase pre-treatment on SPC, replacing 60% of FM, did not improve growth performance in SW Atlantic salmon [98]. The addition of FOS to diets has been observed to improve growth performance of Rainbow trout (Oncorhynchus mykiss) [99]; however, when FOS was added to FM diets for Atlantic salmon, no effect on growth performance was observed [42].

Differences in gut microbiomes between fish fed different diets have been observed in many studies with salmonids, comparing FM and SBM diets [30,32] as well as different treatments of conventional protein sources [18]. Other studies have found that FM-based diets can support greater diversity and community evenness in fish gut microbiomes compared with other protein sources [100,101]. The findings of the present study that USP-fed fish had comparable gut microbiome diversity to US-fed fish, and that ETS-fed fish had lower gut microbiome diversity than all other diets were similar to results for white sea bream (*Diplodus sargus*) fed FOS [102]. However, other existing research has reported greater gut microbiome diversity when prebiotics are added to the diet of juvenile hybrid Tilapia (*Oreochromis niloticus* ♀ × *Oreochromis aureus* ♂) [103]. Existing research indicates that the positive effects of diet treatments on the fish gut microbiome, may be greater with increased exposure period, suggesting a longer term study would be required [104]. It has also been noted that diets with SBM, which contains natural oligosaccharide sources, may mask any beneficial effects of additional prebiotics [102,105]. It will be important to elucidate the impact of low values of gut microbiome diversity and evenness found for ETS-fed fish since high gut microbiome community evenness and diversity values have been associated with greater productivity in aquaculture [106]. In the present study, it may be that this lower gut microbiome diversity and evenness is driven by a community dominance of the lactic acid bacteria (LAB) Enterococcus, a genus that has been associated with growth and immune benefits to fish [107]. However, since the presence of Enterococcus did not exclude other taxa from establishing in the fish gut in this study, this dominance may not have any impact. However, gut microbiome community dominance at such an early life stage should be monitored to ensure that an undesirable dysbiosis does not establish itself in the adult fish [21]. The present study suggests that the application of FOS has quite different impacts than the enzyme-treatment for the gut microbiome when observed at the genus taxonomic level, yet both may be having a prebiotic effect, stimulating the growth of different communities. The increased presence of the LABs in the SBM diets compared with FM is consistent with the existing literature for salmonids [30]. This increase in LABs due to the addition of prebiotic ingredients has also been reported in the common carp (*Cyprinus carpio*) [108]. LABs are of interest to aquaculture as they have been associated with improved digestive function, improved gut health, and disease resistance [38,109,110]. In Rainbow trout, a greater presence of Clostridium_senu_stricto_7, Clostridium_senu_stricto_18 and Peptostreptococcus in the gut has been associated with faster growing individuals, a finding that was also observed in this trial for the fastest growing FM-fed fish [111]. These bacteria have been linked to fermentation of different amino acids [112], suggesting their growth may be facilitated by amino acids present in marine proteins that are not present or less present in fish fed plant proteins. There is a prevalence of Hathewaya and Anaerosalibacter in FM-fed fish compared with all soy-fed fish in this study. These taxa have been found in the gut microbiome of FM-fed Chinook salmon (Oncorhynchus tshawytscha), suggesting an association with this marine protein source [113]. There is a need for future research to address the functional role of LABs in Atlantic salmon and to determine which taxa are associated with growth performance and gut health benefits and thereby improve the application of feed additives and dietary treatments.

The present study is the first to investigate Atlantic salmon behavior traits relevant to welfare and consider if these traits are affected by diets containing different protein sources, and protein treatments. The elicited light stress used in the present study did not appear to produce a response in the fish tested, suggesting the need for a different stress source. It is possible that the fish may have been habituated to light stress as they were kept under continuous light [114]. It is also possible that broodstock selection and domestication of Atlantic salmon genetic lines could have already been selected for more proactive coping styles, reducing responsiveness to the light stress [62,115]. While the traits of exploration and boldness did not differ significantly between the experimental diets, there was a trend that FM fed fish displayed a more proactive coping strategy, i.e., higher boldness and exploration values than fish fed any of the plant-based diets. For ETS-fed fish, there was an indication of more proactive-type behavior compared with US-fed fish. This is consistent with results from juvenile Rainbow trout, where fish fed a plant-based diet, exhibited an increase in apathetic behavior and an increased stress response (both traits of reactive coping styles) [53] compared with fish fed a marine diet [15]. Furthermore, juvenile Senegalese sole (*Solea senegalensis*) fed diets enriched with fish oil from cod liver were more proactive compared with fish fed vegetable oil from linseed, soybean and olive [116]. 

The repeatability of the trait of boldness was greatest in fish fed the FM diet, great enough to indicate the existence of a personality trait which might suggest that fish fed this marine protein diet may have a greater suitability for the conditions found in aquaculture. Conversely, high repeatability of the trait of exploration was shown for fish fed all experimental diets. This could indicate a lesser impact of diet on this trait, i.e., that it is a highly canalized trait, which would be important to elucidate for behavior as a monitoring tool in aquaculture [117]. Existing studies have indicated that more proactive individuals display a higher degree of repeatability for behavior traits, which was also the case in the present study. In the present study, there was also a positive trend between growth performance and the trait exploration in the fish fed the FM diet, although not in any other experimental diet. Previous studies suggest this may be due to a greater level of fitness found in proactive fish compared with reactive fish, e.g., proactive fish have been associated with higher reproductive success in gilthead seabream (*Sparus aurata*) [67], higher growth rates for salmon [118], faster feeding recovery after stress [119] and lesser sensitivity to environmental stress [120] for Rainbow trout, all of which may make them better suited and capable of greater welfare in the culture conditions in fish farms [115]. Repeatability of both traits of boldness and exploration were high for the ETS-fed fish, whereas the fish fed the US diet (and the USP diet to a lesser extent) showed no evidence of any pronounced personality trait. Future studies should elucidate the link between personality trait display, diet and welfare in aquaculture since higher levels of repeatability of behavioral traits may indicate consistent freedom to express natural behavior, which is one of the determinants of good welfare condition [65,115].

In the present study, a clear behavioral syndrome was detected for fish fed the ETS diet, but not for fish fed any of the other experimental diets. This may indicate that the fish fed the ETS diet were experiencing different conditions or pressures than the other fish [58]. It is possible that the domestication of salmon may have limited the presence of behavioral syndromes. In a study of urban song sparrows, no correlation between traits of boldness and aggression could be found, whereas there was a correlation, and thus a behavior syndrome, in their wild counterparts [121]. It is also possible that the behavior of fish in this study was impacted by the culture conditions which may have selected for more plastic behavior or showing adapted behaviors in response to farm stimulus [58]. There is a need for further research attention on the impact of dietary proteins and treatments on behavior and welfare in farmed Atlantic salmon to support the optimization of existing and new protein sources. This study also highlights the need to address knowledge gaps regarding the effects of domestication on salmon behavior and, therefore, welfare in intensive farms. This will help inform selection programs for genetic lines with the best-suited coping styles for culture [122,123].

The present study did not directly analyze the link between individual growth, gut microbiome and behavior; however, this study did identify differences in growth performance in fish fed different experimental diets, differences in gut microbiome between fish fed experimental diets and different trends, although not significant differences, in behavioral traits of fish fed different diets. Collectively, these results suggest that fish fed the FM diet grew better than US-fed fish but were comparable with fish fed treated soy diets (ETS, USP). Fish fed the FM diet had a distinct gut microbiome compared with those fed plant diets (US, ETS, USP). The fish fed the US diet displayed a trend toward being the least bold and having lower repeatability of the trait of boldness, which could lead to reduced welfare, trends that were less pronounced for the treated ETS and USP diets. Previous studies have shown that manipulating the fish gut microbiome can modulate behavior via the gut–brain axis [47], and a recent study on farmed Arctic charr (*Salvelinus alpinus*) [124] showed evidence of bacterial strains influencing the response to stress and growth, where fish fed with *E. thailandicus* 04-394 and *L. brevis* ISCAR-07433 displayed an increase in motility and slower growth, which can be interpreted as a lower stress coping ability. Selecting for personality has been identified as a valuable tool to reduce chronic stress in captive fish, where bolder individuals are usually more resistant to chronic stress [125]. It will be important that future studies take a multi-disciplinary approach to holistically investigate the potential direct links between the collective differences in measures observed in the present study to improve multiple aspects of feed, productivity and welfare and elucidate the gut–brain axis in salmon. Such studies will be essential to the aquaculture sector as they will optimize the suitability of aquaculture feeds, improve broodstock selection, support better welfare monitoring and improve the immune resilience of farmed fish.

## 5. Conclusions

The present results of this multi-disciplinary study show that fish fed marine protein (FM) diets display better growth performance than fish fed soybean meal diets (SBM, US), but that two different treatments of SBM (enzyme pre-treatment (ETS) and addition of a prebiotic (USP)) supported improved fish growth performance to levels comparable with marine protein-fed fish. Fish fed marine protein diets displayed trends towards a more proactive coping style (bolder behavior and more repeatable measures of boldness) than SBM-fed fish, but fish fed treated ETS and USP displayed behavior more similar to FM-fed fish. Distinct fish gut microbiomes were observed between different experimental diets, with lower diversity and evenness in plant-based diets (US, ETS, USP) compared with FM-fed fish. However, for ETS-fed fish, this low diversity was driven by a dominance of the lactic acid bacteria (LAB), Enterococcus. This study demonstrates for the first time the impact of SBM on juvenile Atlantic salmon behavior traits, showing a possible trend for plant-based diets to increase reactive coping styles, which may have adverse consequences for the welfare of fish in intensive farm systems. The results also indicate that the SBM treatments may provide some amelioration that improves the application of plant-based proteins for salmon aquaculture, and these treatments warrant further investigation. Future research should investigate the functional and direct links between protein sources, composition of the gut microbiome and their influence on behavior traits, with a focus on the gut–brain axis, in order to support the salmon aquaculture sector.

## Figures and Tables

**Figure 1 animals-13-02591-f001:**
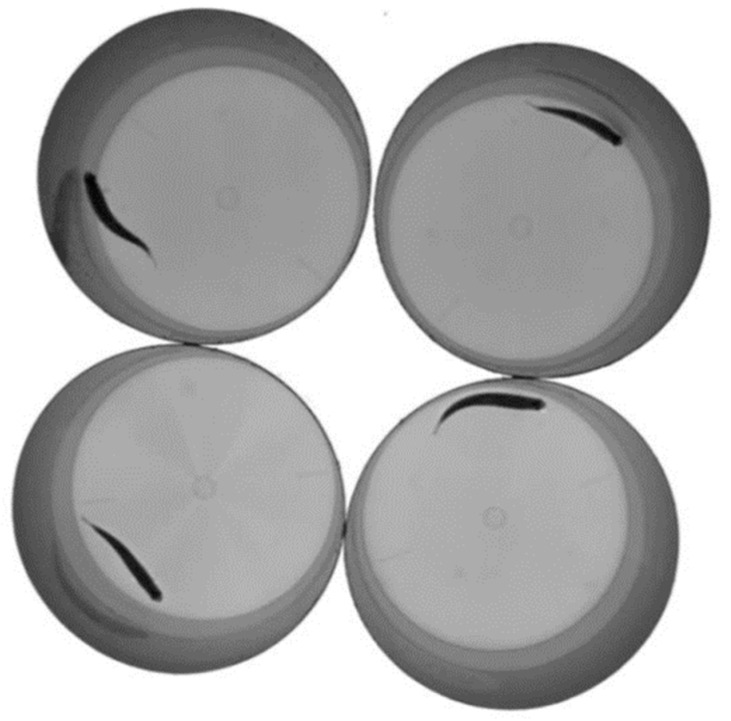
Swimming test arenas used for observation of behavioral traits of exploration and stress response.

**Figure 2 animals-13-02591-f002:**
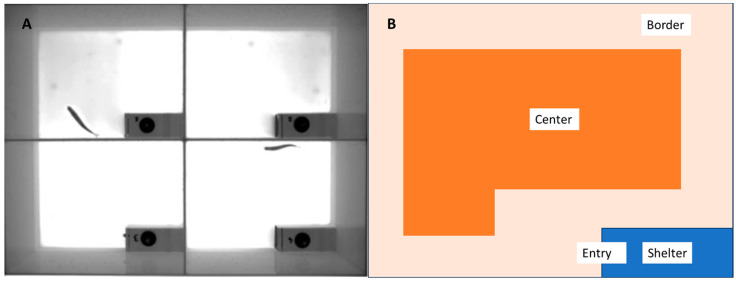
(**A**) Open-field test arenas used for observation of the behavioral trait of boldness. (**B**) An arena with four virtual zones outlined in the EthoVision XT software (Shelter, Entry, Border, and Centre).

**Figure 3 animals-13-02591-f003:**
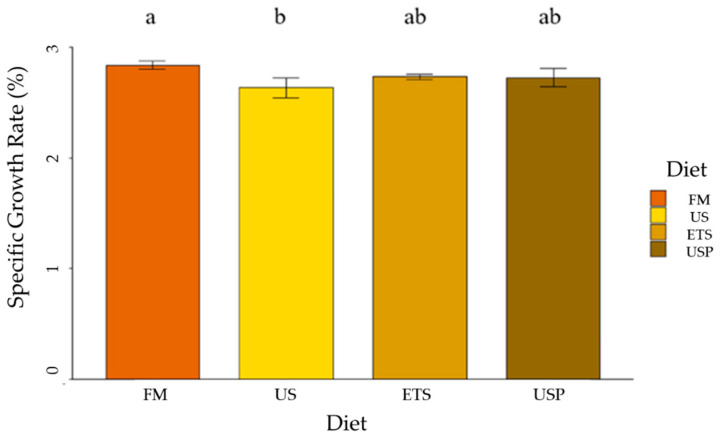
Individual specific growth rate (%) for each diet showing the mean ± SD. Differing lower case letters above the bars indicate significant differences in SGR % (Tukey post hoc test, *p* < 0.05). FM is fish meal-based control diet, US is an untreated soybean meal diet, ETS is an enzyme-treated soybean meal diet, and USP is a diet of untreated soybean meal with fructose oligosaccharide.

**Figure 4 animals-13-02591-f004:**
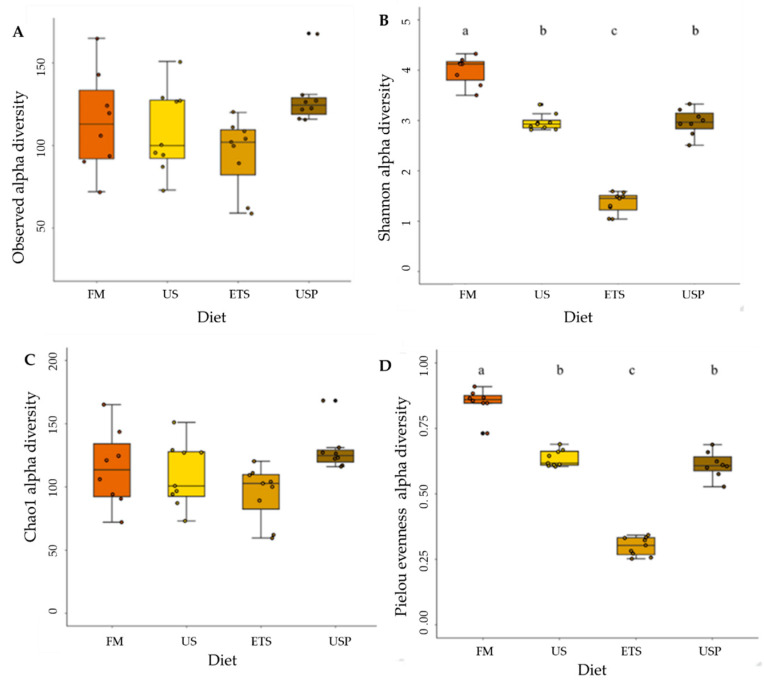
Alpha diversity measures of fish fed different diets: (**A**) observed richness of ASVs, (**B**) Shannon diversity, (**C**) Chao1 diversity, (**D**) Pielou´s evenness. Different letters above the bars indicate significant differences (Tukey post hoc test, *p* < 0.05). FM is fish meal-based control diet, US is an untreated soybean meal diet, ETS is an enzyme-treated soybean meal diet, and USP is a diet of untreated soybean meal with fructose oligosaccharide.

**Figure 5 animals-13-02591-f005:**
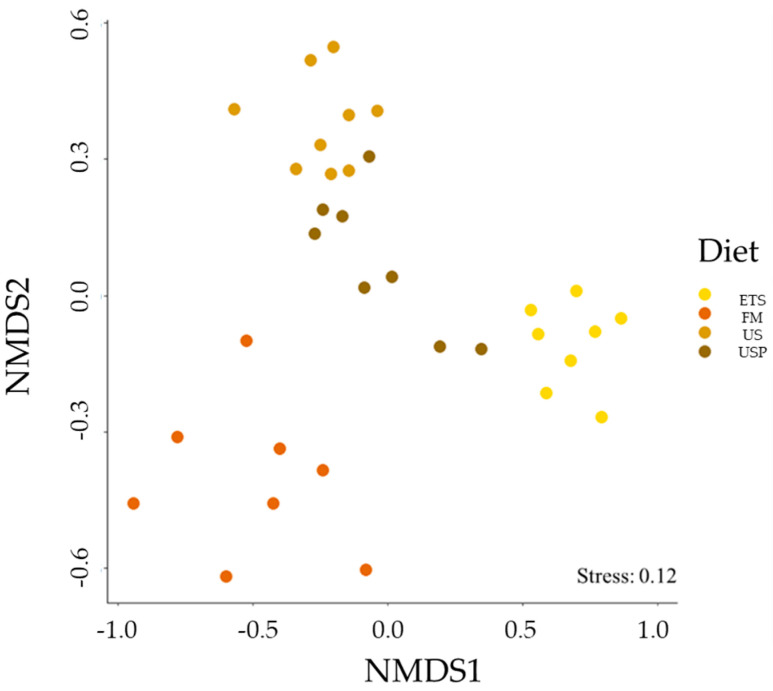
NMDS plot of individual fish gut microbiome composition for each diet. FM is a meal-based control diet, US is an untreated soybean meal diet, ETS is an enzyme-treated soybean meal diet, and USP is a diet of untreated soybean meal with fructose oligosaccharide.

**Figure 6 animals-13-02591-f006:**
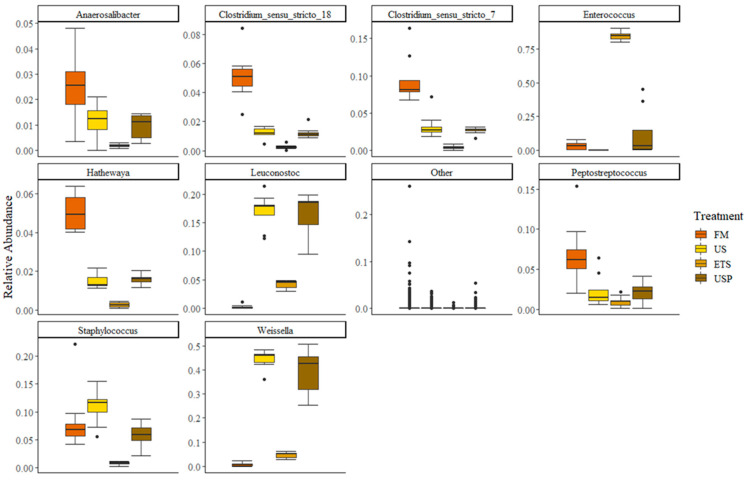
Relative abundance of genera that were present at >1% in the gut microbiome of fish fed each diet, where genera present at <1% are represented by the category “Other”. FM is a fish meal-based control diet, US is an untreated soybean meal diet, ETS is an enzyme-treated soybean meal diet, and USP is a diet of untreated soybean meal with fructose oligosaccharide.

**Figure 7 animals-13-02591-f007:**
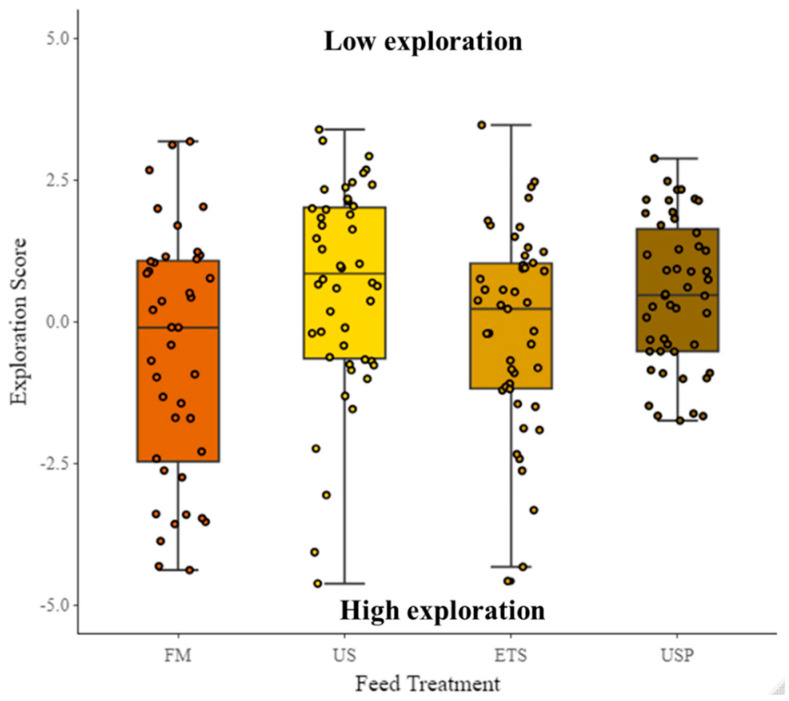
Distance travelled by fish fed different diets. TotDis was used as a proxy for the behavioral trait of exploration. FM are fish fed a fish-meal-based control diet; US are fish fed an untreated soybean meal diet; ETS are fish fed an enzyme-treated soybean meal diet; and USP is a diet of untreated soybean meal with fructose oligosaccharide.

**Figure 8 animals-13-02591-f008:**
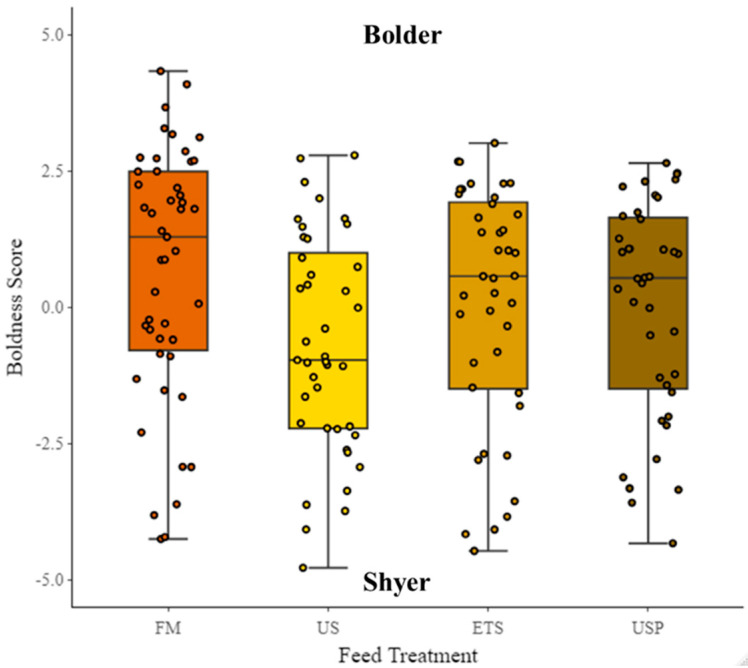
The trait of boldness of fish fed different diets. The lower values indicate shyer individuals and higher values indicate bolder individuals. FM are fish fed fish meal-based control diet; US are fish fed an untreated soybean meal diet; ETS are fish fed an enzyme-treated soybean meal diet; and USP is a diet of untreated soybean meal with fructose oligosaccharide.

**Figure 9 animals-13-02591-f009:**
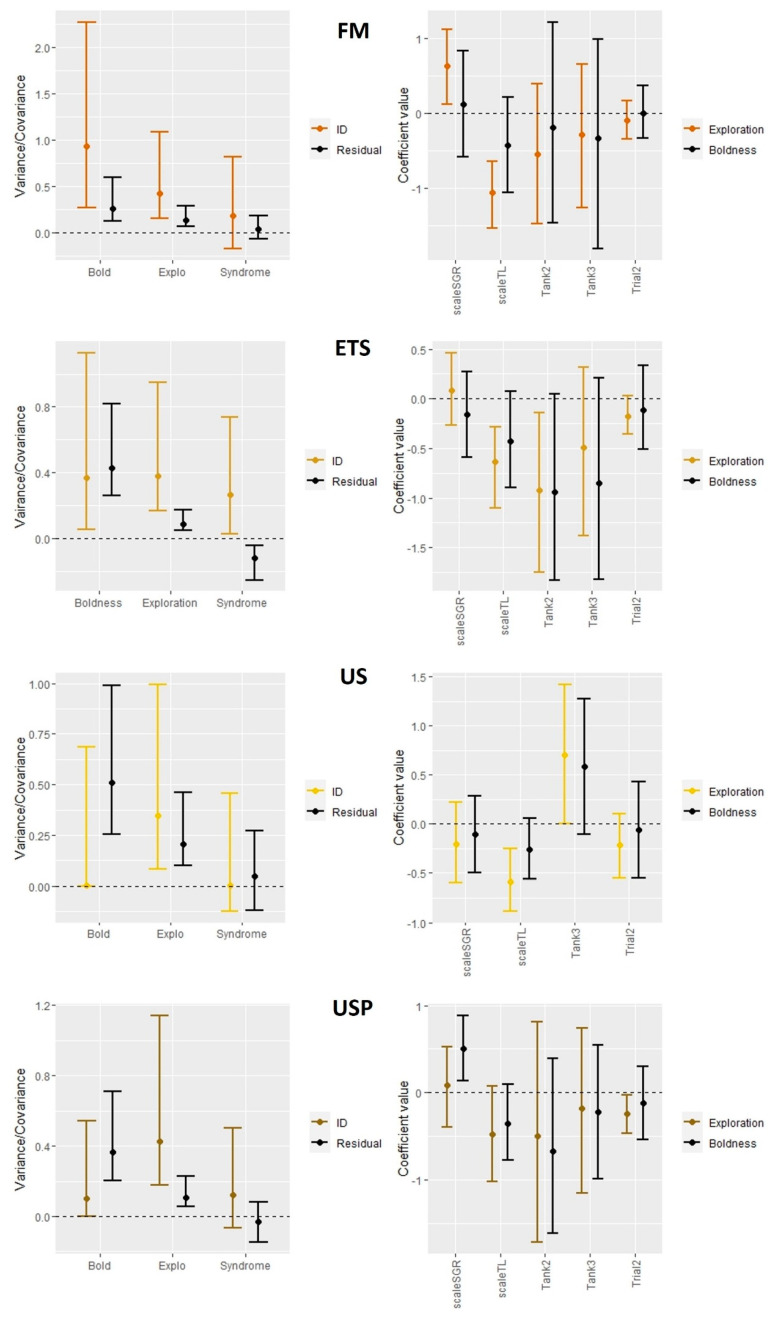
The left columns of the graphs show posterior modes and their corresponding uncertainty intervals for the variances and pairwise covariance of boldness and exploration for each diet treatment, on both the ID level and residual level. The black dashed line indicates zero. The right columns of the graphs show posterior modes and their corresponding 95% uncertainty intervals for the fixed effects SGR, TL, Tank and Trial. TL, SGR and Trial. The dashed line indicates zero. Since the non-categorial fixed effects were z-scored, the coefficient values are presented in units of standard deviation. FM are fish fed a fish meal-based control diet, US are fish fed an untreated soybean meal diet, ETS are fish fed an enzyme-treated soybean meal diet, and USP are fish fed a diet of untreated soybean meal with fructose oligosaccharide.

**Table 1 animals-13-02591-t001:** Chemical composition of the protein sources used in feed treatments in this study.

Composition (g kg^−1^)	Fish Meal	Wheat Gluten Meal	Enzyme-Treated Soy	Untreated Soy (1)	Untreated Soy (2)
Dry matter	909	929	925	926	809
Crude protein	659	742	482	487	479
Crude lipid	107	16	5	9	4
Ash	139	11	73	68	62
**Essential Amino Acids (g kg^−1^)**					
Arginine	40.3	25.0	35.7	34.7	35.1
Histidine	15.3	14.0	13.0	12.4	12.4
Isoleucine	26.0	24.5	22.7	21.4	22.0
Leucine	47.4	48.3	38.6	36.6	36.9
Lysine	51.2	11.4	31.8	30.0	29.7
Methionine	16.8	11.8	6.2	5.9	6.5
Phenylalanine	24.8	35.7	25.8	24.5	26.0
Threonine	28.3	17.8	21.6	20.6	21.8
Valine	32.8	27.4	23.9	22.7	22.5
**Non-Essential Amino Acids (g kg^−1^)**					
Alanine	41.0	18.1	21.8	20.9	21.2
Aspartic acid	59.6	21.9	57.7	55.9	56.0
Glycine	43.6	23.3	21.2	20.5	20.3
Glutamic acid	83.9	260.0	93.6	88.3	87.9
Cystein + cysteine	5.8	15.5	6.4	6.4	6.5
Tyrosine	19.2	22.9	18.1	17.6	18.1
Proline	28.7	86.9	25.2	25.0	23.8

**Table 2 animals-13-02591-t002:** Feed formulation and chemical composition of feed treatments in this study.

Diet Formulation (g kg^−1^)	FM	ETS	US	USP
Fish meal ^a^	702.5	422.6	420.1	420.1
Pre-gelatinised wheat ^b^	189.8	87.8	90.9	90.9
Vitamin–mineral premix ^c^	10.0	10.0	10.0	10.0
Fish oil ^a^	97.8	129.7	128.9	128.9
Wheat gluten meal	0.00	100.0	100.0	100.0
Enzyme-treated soybean meal	0.00	250.0	0.00	0.00
Untreated soybean meal (1)	0.00	0.00	250.0	0.00
Untreated soybean meal (2)	0.00	0.00	0.00	250.0
**Analysed content (g kg^−1^)**				
Dry matter	954	918	934	952
Crude protein	505	485	493	501
Crude lipid	175	170	172	175
Ash	120	93	95	83

^a^ Laxá hf, Iceland. ^b^ Emmelev A/S, Denmark. ^c^ Laxa salmon premix 2006, Trouw Nutrition, Holland.

**Table 3 animals-13-02591-t003:** Repeatability estimates (R) for fish fed different diets. FM are fish fed a fish meal-based control diet, US are fish fed an untreated soybean meal diet, ETS are fish fed an enzyme-treated soybean meal diet, and USP is a diet of untreated soybean meal with fructose oligosaccharide diet. Exploration was assessed using a swimming test and boldness was assessed using an open-field test. CI represents the confidence interval of R.

Diet	Trait	R	CI
FM	Boldness	0.84	[0.49–0.94]
	Exploration	0.82	[0.52–0.94]
ETS	Boldness	0.49	[0.16–0.77]
	Exploration	0.84	[0.64–0.94]
US	Boldness	0.00	[0.00–0.62]
	Exploration	0.72	[0.30–0.89]
USP	Boldness	0.30	[0.00–0.61]
	Exploration	0.85	[0.60–0.94]

## Data Availability

Gut microbiome data is available as a BioProject on the Sequence Read Archive.

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
