# Peer review of "Enhancement of Soybean Meal Alters Gut Microbiome and Influences Behavior of Farmed Atlantic Salmon (Salmo salar)"

_animals, 2023, doi:10.3390/ani13162591_

Round 1

Reviewer 1 Report

The present work assesses the effect of different treatments on soybean meal with the objective of improving its overall nutritional profile for Atlantic salmon fry.

The purpose of the study is interesting and valuable, however some changes need to be addressed, as well as clarification in some points of the manuscript.

line 82 - "the metabolism" appears twice

The introduction section is very long, specifically the part about behaviour can be synthesized.

line 151 - this sentence is confusing. Firstly refers to hatched Atlantic salmon and later to eyed eggs, eyed eggs are still not hatched.

line 186 - please provide a reference, or include the amount of feed required for this development stage.

Did authors consider evaluating diets' acceptability or voluntary feed intake? Might feed intake change due to the different diet composition, thus affecting all the parameters under study?

What is the reason for fasting 12h before microbiome fasting? Fasting modulates gut bacteria composition.

Figure 3: Delete "average" from Y axis title and include in the caption: mean ± SD (or SEM), as correspond. Apply this for all figures.

Globally, for all graphs: make bigger the axis information and legend (numbers and axis titles). Delete frames from graphs because sometimes they overlap (eg. figure 4)

Regarding microbiome composition results, please be consistent in the way the results are expressed. For genera, the composition is expressed as relative abundance (%), while for phylum is not. Please, show both in the same way. 

Include the statistical results in figure 6, as made in figure 4. Include also in the caption that the genera represented are those above 1% of relative abundance.

Results regarding behaviour are confusing and hard to follow, please simplify and/or change the way they are shown.

The authors discuss trends, mostly in behaviour results, that are not supported by statistics. As well as results don't support any conclusion or discussion about the link between microbiome and behavior. Thus, the discussion seems to be a little speculative. 

English must be deeply revised.

Author Response

English language has been reviewed and improved throughout.

Reviewer 2 Report

This manuscript entitled “Enhancement of soybean meal alters gut microbiome and influences behavior of farmed Atlantic salmon (Salmo salar)” mainly compared the detailed impacts of fish meal (FM), soybean meal (SBM), and two targeted enhancements of SBM, enzyme pre-treatment (ETS) and addition of fructose oligosaccharide (USP), on growth, gut microbiome, and related behavior trait in Atlantic salmon (Salmo salar). It will provide the necessary foundation to better understand the advantage of enzyme treatment and prebiotics for improving SBM application and the beneficial actions of probiotic bacteria on the host fishes including Atlantic salmon and develop an optimal plant-sourced feed (especially soy-based diet) for practical application in aquaculture.

In this case, the paper should be proofread throughout for language to eliminate several spelling errors.

Major comments:

1. In “2.1. Experimental animals and study design” section (Line 156), what feed were used in Atlantic salmon during the acclimation period? Commercial feed or other feed? Please offer more information.

2. Regarding the tables, data or information in three-line table are used extensively in the academic papers due to the simple structure and convenience in composition. The format of tables (Table 1-3) in this manuscript should be re-organized.

3. The bar chart, error bars and lowercase letters in Figure 3 did not have the full description in the figure legend (Line 420-424). Please expand the figure legend to provide more information on this figure. Also, there were similar errors in the other figures of this manuscript.

4. According to the statement in Line 419, no significant TotDis (proxy for exploration) was found among feed treatments. But the error bars in Figure 7 were large. Did the huge intra-group errors lead to no differences in TotDis among 4 dietary groups? Please explain it.

Minor comments:

1.Check the symbols or codes for volume unit in this study according to the information to related guides. For example, in Line 272, please replace "ml" and "μl" with "mL" and "μL", respectively. There were similar errors in the other part of this manuscript.

2. In Line 229-230, short notation could be used to indicate the measurement unit of time. For instance, "s" is used to represent "second". There were similar errors in the other part of this manuscript. Please revise it.

Other errors were presented in the PDF file.

Therefore, this manuscript will be reconsidered after minor revision.

This manuscript entitled “Enhancement of soybean meal alters gut microbiome and influences behavior of farmed Atlantic salmon (Salmo salar)” mainly compared the detailed impacts of fish meal (FM), soybean meal (SBM), and two targeted enhancements of SBM, enzyme pre-treatment (ETS) and addition of fructose oligosaccharide (USP), on growth, gut microbiome, and related behavior trait in Atlantic salmon (Salmo salar). It will provide the necessary foundation to better understand the advantage of enzyme treatment and prebiotics for improving SBM application and the beneficial actions of probiotic bacteria on the host fishes including Atlantic salmon and develop an optimal plant-sourced feed (especially soy-based diet) for practical application in aquaculture.

In this case, the paper should be proofread throughout for language to eliminate several spelling errors.

Major comments:

1. In “2.1. Experimental animals and study design” section (Line 156), what feed were used in Atlantic salmon during the acclimation period? Commercial feed or other feed? Please offer more information.

2. Regarding the tables, data or information in three-line table are used extensively in the academic papers due to the simple structure and convenience in composition. The format of tables (Table 1-3) in this manuscript should be re-organized.

3. The bar chart, error bars and lowercase letters in Figure 3 did not have the full description in the figure legend (Line 420-424). Please expand the figure legend to provide more information on this figure. Also, there were similar errors in the other figures of this manuscript.

4. According to the statement in Line 419, no significant TotDis (proxy for exploration) was found among feed treatments. But the error bars in Figure 7 were large. Did the huge intra-group errors lead to no differences in TotDis among 4 dietary groups? Please explain it.

Minor comments:

1.Check the symbols or codes for volume unit in this study according to the information to related guides. For example, in Line 272, please replace "ml" and "μl" with "mL" and "μL", respectively. There were similar errors in the other part of this manuscript.

2. In Line 229-230, short notation could be used to indicate the measurement unit of time. For instance, "s" is used to represent "second". There were similar errors in the other part of this manuscript. Please revise it.

Other errors were presented in the PDF file.

Therefore, this manuscript will be reconsidered after minor revision.

Reviewer 3 Report

Analysing the swimming behavior of fish could be an adequate non-invasive tool of welfare indicator. Individual testing of salmon parr swimming behavior can provide useful information, however group behavior could be even better for further analysis.

Most of the methods are well described with sufficient references, however in some cases more detail are necessary. 

Fish keeping conditions: please add more information was it flow-trough or RAS? Presenting the basic water parameters can be usefull (different forms of Nitrogen, pH etc..)

Line 187 "on the feed requirements for this developmental stage" please add reference

Line 209 to 223 please add references for application of 2-phenoxy ethanol for safe and lethal dose of application

table 2 Analysed content (g kg-1) the sign of % should be deleted

the results are well described and they agree with the literature FM vs US vs. ETS and USP

Round 2

Reviewer 1 Report

Thank you for the work on addressing and clarifying all comments.

Just two issues for finishing the review process:

Please, change "study diets" to "experimental diets". The first term is not frequently used.

Since microbiota sampling was performed after a period of fasting, and assuming that the digestive tract was empty at sampling, authors only sequenced mucosa-associated microbiota and not luminal microbiota (found in gut content/ feces). This should be stated and considered in the manuscript. 
